# Emissions Control of Hydrochloric and Fluorhydric Acid in cement Factories from Romania

**DOI:** 10.3390/ijerph17031019

**Published:** 2020-02-06

**Authors:** Gheorghe Voicu, Cristian Ciobanu, Irina Aura Istrate, Paula Tudor

**Affiliations:** 1Department of Biotechnical System, Faculty of Biotechnical Systems Engineering, University Politehnica of Bucharest, Spaiul Independentei 313, Sector 6, RO-060042, 010164 Bucharest, Romania; ghvoicu_2005@yahoo.com (G.V.); ciobanu_77@yahoo.com (C.C.); 2Ceprocim Sa–Strada Preciziei 6, RO-062203, 010164 Bucharest, Romania; 3Department of Management, Faculty of Entrepreuneurship Business Engineering and Management, University Politehnica of Bucharest, Spaiul Independentei 313, Sector 6, RO-060042, 010164 Bucharest, Romania; paulavoicu85@yahoo.com

**Keywords:** cement factory, clinker kiln, pollutant emissions, hydrochloric acid, hydrofluoric acid

## Abstract

From the available statistical data, cement factories co-process a range of over 100 types of waste (sorted both industrial and household) being authorized for their use as combustion components in clinker ovens. Therefore, the level of emissions is different depending on the type of fuels and waste used. The amount of industrial and municipal co-processed waste in the Romanian cement industry from 2004 to 2013 was about 1,500,000 tons, the equivalent of municipal waste generated in a year for 18 cities with over 250,000 inhabitants. The objective of this paper was to evaluate the emission level of hydrochloric acid (HCl) and hydrofluoric acid (HF) at the clinker kilns at two cement factories in Romania for different annual time intervals and to do a comparative analysis, to estimate their compliance with legislation in force. The measurements results showed average emission levels of about 0.578 mg/Nm^3^ for HCl and about 0.100 mg/Nm^3^ for HF, in the first hours of the evening, but decreased at the beginning of the third tour, at about 0.385 mg/Nm^3^ for HCl, respectively, to about 0.085 mg/Nm^3^ for HF. The evolution of HCl and HF emission levels during the last 4 years showed a variable distribution of these acids.

## 1. Introduction

Spatial urban growth, land use changes, decreased carbon sinks, and increased energy consumption in buildings and transport are major challenges in reducing emissions and in regional and urban development planning. The decision-makers need to be aware of the growth of urban settlements and the modification of the land use, associated with an increase in the level of spatial emissions for an adaptive policy at relatively short intervals. Scenarios can be built, and specific programs can be used to estimate population growth and employment to estimate the level of emissions, mainly those with a greenhouse effect, and mitigate future emissions [1,2].

Moving to a low-carbon economy is becoming more critical and difficult, especially for developing countries. In the concrete industry, emissions during the production of raw materials and their transport, from suppliers to users, are high in terms of greenhouse emissions level. Raw material suppliers and companies that produce or use cement or concrete must come up with plans to reduce greenhouse gas emissions using alternative types of cement, substitutes for alternative aggregates that reduce water consumption, and use recycled industrial waste, encouraging users to search for suppliers nearby. Based on the study presented by Akan et al. (2017), it will be difficult for developing countries to meet the dual objectives of economic prosperity and environmental sustainability [3]. 

Recent estimates show that cement accounts for about 5–10% of global anthropogenic carbon dioxide (CO_2_) emissions, with more than half of them being related to clinker production due to the use of fossil fuels and direct clinkering emissions. With all the worldwide emissions reductions in cement plants, it seems that Portland cement (mainly composed by calcium silicate minerals) tends to have greater global warming impacts than when cement in the mix (with natural pozzolan or blast furnace slag) is produced [4].

In their work, Geng et al. (2019) state that most clinkers contain small amounts of carbonate that are not completely decomposed and small amounts of fossil fuels that are not completely burned. They propose that the process-related emission factor be calculated based on the decomposition ratio of carbonate (inorganic carbon) and the emission factor related to combustion be determined by multiplying the total organic carbon (TOC) by its combustion ratio [5].

The best available techniques should use conventional fuels with low chlorine and fluorine content, respectively, limiting their content for any waste to be used as a fuel in a combustion kiln in order to reduce HCl and HF emissions from gases resulted from the combustion processes of the kilns in cement factory [6]. It is important to remember that to obtain a ton of clinker, about 3000–6500 MJ of thermal energy is required, and the current trend in the field of cement production is the focus on low energy cements, the use of waste in cement production, and the associated reduction of CO_2_ emissions [7].

Fluorine is a common element in the earth’s crust, and its salts are naturally present in soil, rocks and water throughout the world. They are used in many industrial processes, but also result as emissions from other processes, either separately or in combination with other types of emissions [8].

Fluoride appears in small quantities in plant matter, through soil absorption, but also in animals, through food and water consumption. In animals, the retained fluoride is mainly concentrated in bones and teeth. In excessive amounts, fluoride can cause chronic poisoning in animals, called fluorosis, even if some animals (such as cattle) have a higher tolerance for fluoride than others.

Excessive systemic exposure to fluoride can lead to disorders of bone homeostasis (skeletal fluorosis) and enamel development (dental fluorosis/enamel). Mitogen-activated protein kinases (MAPKs) signaling pathways allow cells to respond to myriad extracellular stimuli. Fluorides mediate their actions through MAPK signaling pathways, leading to changes in gene expression, cell stress, and even cell death [8].

Emissions of fluoride pollutants into the atmosphere generally come from industrial processes, either the fluorine compounds are produced directly, are used as process catalysts, or may result as impurities in the process materials.

The literature shows that the primary mechanism of fluoride release in high-temperature processes is pyrohydrolysis, which results in HF formation [9]. In some cases, volatile metal fluorides may form, but this is a less critical mechanism than pyrohydrolysis. Thus, fluoride or HF emissions can occur in calcining ironstone or topaz; the making of steel, brick or glass; the manufacturing of calcium metaphosphate and defluorination; or the manufacture of superphosphate [9].

Emissions of hydrofluoric acid into the air from foliar fluorides, in the vicinity of traditional brick kilns, have affected mango, apricot, and plum orchards in Asia, due to increased concentrations due to the malfunctioning of ovens [10,11].

Fluoride emissions from an aluminum factory in Greece caused toxicity to natural vegetation in the surrounding area, some plant species accumulating high levels of fluoride in the leaves, which resulted in acute necrosis and/or chlorosis. Average levels of fluoride in the vegetation can range from 257.2 to 621.2 ppm in severely affected areas (zone I) and from 64.1 to 144.3 ppm in slightly degraded areas (zone III). In soil, total fluoride concentrations ranged from 297.6 to 823.5 ppm; control levels ranged from 95.3 to 108.6 ppm [12].

The cement industry is an essential generator of pollutant emissions both in the atmosphere and in surface waters, not only through greenhouse gases (CO_2_) and primary pollution (NOx, PM, SO_2_) but also through numerous atmospheric pollutants (HCl, NMVOC, PCDD/Fs, PAHs, and fluorides), as well as heavy metal emissions (As, Cd, Cr, Hg, Ni, Pb, Zn, and Cu), which are often neglected [13].

Analyzing seven types of alternative fuels in clinker ovens (Richards et al., 2015), it was found that the waste oil, wood chips and plastic, waste solvents, and tire-derived fuels are favorable in reducing target emissions values to normal operations, while providing the required energy demands for clinker production and ensuring the energy demand for clinker production. Monitored pollutants refer to CO_2_, CO, NOx, SOx emissions, hydrogen halides and halogens, and total volatile organic compounds.

The key parameters of the clinker obtaining process, which contribute to the significant reduction of contaminants, include: precalciner and kiln fuel firing rate and residence time; preheater and precalciner gas and material temperature; rotary kiln flame temperature; fuel–air ratio and percentage of excess oxygen; and the rate of meal feed and clinker produced [14].

According to Emission Monitoring and Reporting (EMR) requirements, in Romania, cement factories must measure hydrochloric acid (HCl), ammonia (NH_3_), benzene (C_6_H_6_), heavy metals, and dioxins/furans and continuously calibrate emission monitoring equipment (EMC) for all species measured at least once a year. Emissions, such as dust, NOx, SO_2_, VOC (volatile organic compounds), O_2_, and humidity, are part of group A emissions; while hydrochloric acid (HCl) and ammonia (NH_3_) are part of group B;a and benzene (C_6_H_6_), heavy metals (HM) and dioxins/furans (PCDD/PCDF) re from group C emissions [15,16].

Regarding the fluoride present in the cement factory kilns, 90–95% is bounded in the clinker, and the rest is bound in the powder as a calcium fluoride form stable under the burning process conditions. Due to calcium excess, gaseous emissions of fluoride and hydrofluoric acid are excluded. About 70% of the measured emissions at cement kilns are below the detection limit of approx. 0.05 mg/Nm^3^. The maximum emission is 1 mg/Nm^3^ [15,17].

As for chlorine, it is a powerful oxidant and is very toxic to the human body. No matter what form it reaches in the body, either in the gaseous state or in the liquid state, the effect is the same. Chlorine is a very powerful irritant that contributes to the accumulation of free radicals, having effects on the cells’ health. Chlorine gas emissions can cause eye and respiratory irritation, sore throat, and cough. At a higher level of exposure, symptoms may develop a feeling of tightness in the chest area, in wheezing and/or bronchial spasms. Multiple exposures may result in non-cardiogenic pulmonary edema.

The chlorides are minor additional constituents of raw materials and fuels in the burning process of clinker in vertical kilns with cyclone. They are released when the fuel is burned, and the flue gas reacts primarily with the alkaline substances in the feedstock forming alkaline chlorides. These compounds, which are initially in the form of vapors, condense into the raw material, especially at temperatures of 700–900 °C, and re-enter the kiln and evaporate again. This cycle in the space of the furnace can lead to agglomerations. In the case of vertical kilns with cyclones, about 70% of the measured emissions for inorganic chlorine compounds are below 2 mg/Nm^3^; in the remaining cases, they are between 2–10 mg/Nm^3^. Chlorine bypass furnace systems may have higher emission ranges [18].

According to Schorcht et al. (2013), the HCl emissions from the clinker ovens of the cement plants in the Member States of the European Union, which are characterized by different replacement percentages of traditional fuels, presented average values of about 4.26 mg/Nm^3^, with a standard deviation of 4.6 mg/Nm^3^ [19].

In the cement factories, the organic composition of the waste is destroyed by combustion producing thermal energy, while the mineral composition is recycled by chemically integrating them into the structure of the clinker. This way, no slag or ash results after the combustion of fuels for clinker production. The burned waste type from the clinker kilns contributes to the number of emissions discharged into the atmosphere. Thus, burning wood chips with plastics and tires can increase chlorine emissions (by 26% and 89%, respectively), [20,21].

To this end, Romanian cement factories have invested over 100 million euros in high-performance equipment for pre-treatment and co-processing of waste.

According to the International Finance Corporation (IFC) Environment, the allowable emission limit values for cement kilns must be: dust from kiln firing:50 mg/Nm^3^, NOx: 600 mg/Nm^3^, SOx (as SO_2_): 400 mg/Nm^3^, hydrogen chloride (HCl): 10 mg/Nm^3^, and hydrogen fluoride (HF): 1 mg/Nm^3^ [22]. Numerous studies have linked atmospheric pollutants to many types of health problems of many-body systems, including the respiratory, cardiovascular, immunological, hematological, neurological, and reproductive/developmental systems [23]. That is why one of the future researches related to this subject will be the study of the human health risk assessment associated with HCl and HF emissions.

The average limits of pollutant emission for measurements with an exposure period of 30 min are as follows: 60 mg/Nm^3^ for hydrochloric acid, 4 mg/Nm^3^ hydrofluoric acid and a minimum of 97% of the measurements of the values are about 10 mg/Nm^3^ in hydrochloric acid and 2 mg/Nm^3^ in hydrofluoric acid [22].

Solid fuels, recovered from municipal waste, used in clinker ovens of cement factories, should have certain characteristics, mainly being established by the CEN/TC343 standard. This standard provides a high calorific value (about 16–18 MJ/kg, although some installations even require higher values); a small amount of chlorine (below 0.5%); and a small amount of mercury (below 10 mg/kg, d.b.). For such fuels, the amount of chlorine measured in the emissions samples of cement plants can be about 2000 mg/kg [24,25].

The use of sewage sludge as an addition to the fuel used in furnaces results in increased emissions of polycyclic aromatic hydrocarbons (PAHs) and heavy metals, in particular high molecular weight PAHs and low volatility heavy metals, such as Cd and Pb, as particles. Over 95% of the total PAHs in the flue gas for cement production are in the gas phase, and the majority of them are low molecular weight PAHs [26].

It is essential to monitor these values because the consequences appear to be quite disastrous. For example, lethal accidents have been reported in workers at different workplace settings following burns of as little as 2.5–8% of their skin surface with highly concentrated HF (>70%), leading to ventricular fibrillation, metabolic acidosis and multiple organ failure [27,28,29]. Although HF is a weak acid (pK = 3.19; dissociation constant = 4 × 10^−4^ a mol/L), dermal absorption may lead to damage of deeper tissue. Dependent on the concentration of HF, skin surface damage and pain might occur in a delayed manner. HF-induced skin damage was investigated in several histological studies in different species [30,31,32,33].

A New Zealand study reported personal air exposures of 75 ppm SO, 25 ppm HCl, and 8 ppm HF in 10 human volunteers who spent 20 min near volcanic vents during a quiet period of the White Island volcano [23,34].

Decreased lung function and skin/eye/nose/throat irritation were noted in many members of a Texas community of 3000 exposed to an HF spill from a nearby oil refinery [23,35].

In order to guarantee the characteristics of waste used as fuels and/or raw materials in a clinker kiln from cement factories and to reduce emissions, the practical techniques consist in the application of quality assurance systems and the control of the relevant parameters of the waste, such as: chlorine, the content of relevant metals (cadmium, mercury, thallium), sulfur content, and total halogen content.

To ensure proper treatment of waste used as fuels and/or raw materials in the furnace, the techniques should raise the temperature to 1100 °C if hazardous waste with a content of more than 1% of substances is co-incinerated. Also, in order to prevent and reduce HCl emissions from the resulted gases of the combustion processes in the clinker oven, the primary techniques consist of the use of raw materials and fuels/waste with low chlorine content.

The addition of mineralizers, such as fluorine, in the raw materials, is a technique for adjusting the quality of the clinker and allows to reduce the temperature in the clinker area. By reducing the combustion temperature, the formation of NOx is also reduced [6].

In cement factories, compounds analyzed with continuous emission measurement equipment must be checked (calibrated) at least once a year to confirm continuous quality measurements, i.e., parallel analysis with a standard reference method or validated method measuring are done to control the emission monitoring equipment (EMC)’s estimated values. 

Of all the components of continuous EMC, some should be specially calibrated, namely nitrogen monoxide (NO), sulfur dioxide (SO_2_), oxygen (O_2_), and water vapor (H_2_O). It is advisable to calibrate and measure the continuously measured compounds such as carbon monoxide (CO), ammonia (NH_3_), hydrochloric acid (HCl), etc. [36,37,38]. The main objective of the paper is the comparative analysis of HCl and HF emissions level at two of the Romanian cement factories, for several consecutive years, to estimate their classification within the allowed limits and to analyze the possibilities of reducing them. The factories are located in two different areas and will be mentioned in this paper as the first factory and the second one. The first one is built in the southern central area of Romania and has a capacity of over 5000 tons of clinker per day and a capacity of co-incineration of waste of 3 t/hour in the clinker oven. The second factory is built in the central western area of Romania and has a capacity of over 4500 t/day clinker and a waste co-incineration capacity of about 10 t/hour at the clinker ovens.

## 2. Materials and Methods 

In order to obtain Portland cement, the raw materials (basic material = limestone, clay, marble), which are finely shredded and the fuel (coal, oil, gas, oil sludge, used tires, various types of previously processed waste) are introduced. This mixture is burned at a temperature of about 1450 °C. In the furnace (Figure 1), the base material and the ash resulting from the combustion of the fuel are transformed into a fluid mass that at the exit is suddenly cooled, solidifying. Then the clinker, thus obtained, is finely shredded together with the addition of gypsum, resulting in Portland cement. Emissions resulting from combustion in the furnace then pass through specific filters, finally being evacuated through the dispersion basket of the factory, whose output is located at a high altitude. The content of the discharged emissions therefore depends to a large extent on the composition of the fuel used and, in particular, the burned waste.

The non-isokinetic method for determining the level of halides and halogens emitted from stationary sources involves several steps. A sample is extracted from the source and passed through a preheated filtration sample and filtered in dilute sulfuric acid and sodium hydroxide solution, which will collect the gaseous halides and halogen. The filter collects particles, including halide salts, but they are not usually collected or analyzed. The halogen has low solubility in acid solution and that is why it will pass into an alkaline solution where it is hydrolyzed to form a proton (H^+^), halide ion, and hypohalosic acid (HClO or HBrO). Sodium thiosulphate is added in excess to the alkaline solution to provide the reaction with hypohalosic acid to form a second halide ion so that the two ions form a gas molecule. To measure halide ions from separate solutions, ion chromatography (IC) is used [15,39].

In the paper, the compounds (HCl, HF) were measured by sampling and analysis according to the standard method using an Advance Cemas FTIR monitoring system. This system is an emission monitoring system for several components simultaneously, such as HCl, HF, NH_3_, CO, NO, NO_2_, SO_2_, H_2_O, CO_2_, O_2_, and volatile organic compounds (VOC), if a flame ionization detector (FID) filter is integrated. The system consists of a TESTO 400 portable instrument for determining the temperature, humidity and dew point in the pipe; a BRAVO M Plus constant flow sampling pump for non-isokinetic sampling of HCl and NH_3_; and a TESTO 350 XL and TESTO 350 gas analyzer for determining the O_2_, NOx, CO, and SO_2_ values in the pipeline [40,41].

The measurement of the active components with infrared is done at high temperatures (180 °C) using an FTIR spectrometer. For the data in the paper, we used a PRECISA analytical balance (for the dust mass collected on the filter) and a DINEX ion chromatograph type ICS 3000.

The simplified scheme of the sampling system is represented in Figure 2.

O_2_ measurement is performed using an electromechanical oxygen sensor (zirconium oxide sample). The total content of volatile organic compounds is measured using an FID filter.

The sample collection system consists of a sample tube (length 500–2500 mm, process temperatures <500 °C), a filtering equipment (heated up to 180 °C), sample gas lines (heated up to at 180 °C, with a length of up to 60 m), a reverse sample filtration module, and an automatic test gas injection system in the sample for checking the battery.

The analyzing gas treatment unit (of the gas sample) has a built-in SS-microporous filter (having the possibility of heating the gas); a module with an air injector pump; connection ports for oxygen and volatile organic compounds; automatic connection; and shut-off for gas calibration or non-supply, pressure, temperature and flow sensors.

For calibration, all FTIR-dependent factors are analyzed using zero-day automated daily recordings. The zero value and the deviation are automatically corrected since absorption of the spectrum is absolute and does not vary. Manual gas calibration verification can be easily performed on the analyzer cell or sample, according to international requirements. For the FTIR spectrometer, the data capture period is 120 s, and the critical optical path length is 6.4 m; O_2_ was measured using an integrated electromechanical cell; the detection limit (LOD) being 0.26 mg/m^3^ (i.e., 0.16 ppm) for HCL and 0.12 mg/m^3^ (i.e. 0.13 ppm) for HF.

The values of the pollutant concentration *C* in (ppm) (dry basis-*C_dry_*) must be converted to (mg/Nm^3^-*C_dry.N_*) (for “normal conditions”: 1013 mbar, 0 °C), using the relation:(1)$$C_{dry.N}\left[\frac{\mathrm{mg}}{{\mathrm{Nm}}^{3}}\right]=fC_{dry}\;\left[\mathrm{ppm}\right]$$
where the conversion factors, *f*, are 1.58 for HCl (which has a molar mass of 35.46 kg/kmol) and 0.89 for HF (which has a molar mass of 20.01 kg/kmol).

The measured values of the pollutant concentration are indicated at a standard temperature of 20°, 25° or another temperature (but not 0 °C), and 1013 mbar (*C_wet_*), “standard conditions”. Also, all measured values (including oxygen concentration) must be converted to gas (*C_dry_*):(2)$$C_{dry}\left[\frac{\mathrm{mg}}{m_{s.\mathrm{dry}}^{3}}\right]=C_{wet}\left[\frac{\mathrm{mg}}{m_{s.\mathrm{wet}}^{3}}\right]\times \frac{100}{\left(100-H_{2}O\left[\%\right]\right)}$$

For the first cement factories analyzed in 4 consecutive years (2015–2018), the fuels used for combustion were: coal 5908 kg/h, waste oil 1800 kg/h, distillation residues 156 kg/h, animal flour 681 kg/h at the main burner, and plastics 1024 kg/h at the secondary burner. 

Three measurements were made every 5 and 10 min, at the dispersion stack of the dry gas effluent, at 140 m above the ground level for the first cement factory and 40 m for the second cement factory. Sampling took 30 min, and measurements were made either on the same day or the following day, on the two branches of the exhaust gas flow (branch A and branch B). These data have been processed for presentation to the factory standard, which provides the conversion from the oxygen reference content to 10% (vol.). The samples presented in the paper were made every year, approximately during the same period of the year (August–September).

The analyzed cement factories have their standard. They present the values measured under different conditions of pressure, temperature, humidity. To be more exact, the data presented in this paper are obtained at a pressure of 1013 mbar, temperature of 0 °C, for dry material and at 10% vol O_2_ (or other reference content). If we use the same values for other conditions, then a conversion is done using a local conversion factor, depending on the atmospheric pressure of the samples, their humidity and temperature, as well as their oxygen content (in volumes). 

The conversion from the oxygen content of the measured sample O_2_ (% vol.–*C_dry.N_*) to a volume of 10% oxygen was done using the relation (*C_Factory_*):(3)$$C_{Factory}=C_{dry.N}\left[\frac{\mathrm{mg}}{{\mathrm{Nm}}^{3}}\right]\times \frac{21-10\left[vol\%\right]}{21-O_{2}\left[vol\%\right]}$$

In order to determine the quantity of the substance emitted in time, it is also necessary to measure the volume of the source that is verified using the relation:(4)$$V_{tot}=\left(0.25\;q+0.27\right)\left(1+\frac{O_{2}}{21-O_{2}}\right)$$
where: *V_tot_* is the specific volume of the gas discharged under normal (standard) conditions (0 °C, 1013 mbar) and dry gas state, in m^3^/kg clinker; *q*-calorific consumption of the oven, in MJ/kg clincher; and O_2_, oxygen content at the measurement point, in% (dry volume).

No relative deviations between the volume of the calculated source and the measured one are higher than ±10%.

## 3. Results and Discussion

The measurement results regarding the HCl and HF emissions, for the first monitored cement factory, are presented in Table 1 and Table 2. These measurements showed, e.g., average emission levels of about 0.578 mg/Nm^3^ for hydrochloric acid and about 0.100 mg/Nm^3^ for hydrofluoric acid, during the first hours of the evening, but decreased at the beginning of the third shift at about 0.385 mg/Nm^3^ for hydrochloric acid, and approximately 0.085 mg/Nm^3^ for hydrochloric acid.

As a general finding, none of the measured values exceeded the limits of the Ordinance on Air Pollution Control (OAPC). According to the OAPC, emissions of chlorine must not exceed 3 mg/m^3^ in installations for the production of chlorine, that being 20 mg/m^3^ for gaseous inorganic chlorine compounds, expressed as hydrogen chloride, in installations for the incineration of municipal and special waste. Also, here the emissions of gaseous inorganic fluorine compounds, expressed as hydrogen fluoride, should not be greater than 2 mg/m^3^. For the kilns of cement factories, the values presented in the IFC [25], mentioned in the introduction, are used. The values indicated and measured with the OPSIS equipment were in the correct range of values of the extraction method (the reference method), and no corrections or adjustments of the EMC (continuous emission monitoring) were required.

The samples used for the analysis were from a higher point of the dispersion stack (located at 140 m height), which is the evacuation area in the atmosphere. In 2018, the speed of the dry gas residual effluent was between 23–23.4 m/s when the cement factory was operating without the raw materials mill and between 19.5–20.3 m/s when the factory had the raw materials mill in operation. Also, the values of the effluent pressure at the discharge were between 92.69–92.72 kPa and between 92.36–92.50 kPa for the same working conditions. The effluent temperature had values between 172–175 °C without the raw materials mill, and between 107–113 °C when the factory operated with the raw materials mill. The calculated volumetric flow for the effluent was about 289–294 m^3^/s and 245–256 m^3^/s for the working conditions mentioned above (without and with the raw materials mill, respectively).

From Table 1 and Table 2, as well as from Figure 3, it is noted, however, that HCl and HF emission levels have increased during the analysis done in the last 2 years, which could be accounted for by the increasing quantities of waste used in the clinker kiln.

For the second cement factory, the obtained values for the samples are presented in Table 3 and Table 4. 

In 2015, the oxygen values in the taken samples were about 9.2% for both rotary kiln feeders, with the conversion to a volume of 10% oxygen being necessary. The volumetric flow, converted to 10% O_2_ (vol.), had values between 38.8–44.7 Nm^3^/s (for branch A) and between 49.4–51.2 Nm^3^/s (for branch B), while the average mass flow rate for HCl emissions was 384.1 g/h for branch A and 68.3 g/h for branch B, and 62.3 g/h (for branch A) and 21.8 g/h (for branch B) for HF emissions. The velocity of the gas effluent at the outlet in the atmosphere of the emission tower was on average 10.9–12.7 m/s for six samples taken at intervals of 35-60 min, at a pressure of 98.7–98.8 kPa and at a temperature from 143–147 °C, when the factory operated without the raw materials mill.

For 2017, the oxygen values in the samples were between 7.69–8.29% for branch A and between 8.05–9.00% for branch B, requiring the conversion to a volume of 10% oxygen. The volumetric flow, converted to 10% O_2_, had values between 38.03–45.48 Nm^3^/s, while the average mass flow for HCl emissions was 46.61 g/h for branch A and 36.42 g/h for branch B, and 7.83 and 7.85 g/h for HF emissions. The velocity of the gas effluent at the outlet in the atmosphere of the dispersion stack was as an average for seven samples around 10.6–13.7 m/s taken at intervals of 35–60 min, at a pressure of 98.25–98.33 kPa, at a temperature from 121–123 °C, when the factory operated with a raw materials mill.

For the year 2018, the oxygen values in the taken samples were between 8.26–8.57% in branch A and between 7.99–9.02% in branch B, requiring the conversion to a volume of 10% oxygen. The volumetric flow rate, converted to 10% O_2_, had values between 37.6–39.5 Nm^3^/s (branch A) and between 51.6–52.3 Nm^3^/s (branch B), while the average mass flow for HCl emissions was 54.3 g/h for branch A and 155.03 g/h for branch B, and 8.63 g/h (at branch A) and 15.46 g/h (at branch B) for HF emissions. The velocity of the gas effluent at the outlet in the atmosphere of the emission stack was on average 11.1–11.5 m/s (branch A) and 14.9–15.0 m/s (branch B), for the six samples taken at 35–60 min intervals., at a pressure of 100.8–101.0 kPa, at temperatures of 130.4–132.0 °C, when the factory operated without raw materials mill.

There are differences between the HCl and HF emissions in the exhaust gases, although each year they do not exceed the limit values recommended by the regulations in force. The differences that arise are due to the quantities of co-incinerated waste and the mixture made for combustion in the clinker ovens. Although the used mixtures are specified in the paper, they differed from period to period and from year to year. The authors did not have data regarding the quantities of burned waste, by category, at the time of the tests, but it is assumed that they were within the limits specified above.

It can be stated, however, that the gaseous emissions level of HCl and HF at the first analyzed cement plant increased in 2017–2018, as compared to 2015–2016, which is probably due to the introduction of co-incineration in clinker ovens of larger quantities of waste and with higher chlorine and fluoride content. In contrast, at the second cement plant, HCl and HF emissions presented higher values (but below the regulated limit) in 2015 in branch A, and show a significant decrease in values in the years 2017–2018.

The chlorine and fluorine content of the co-incinerated waste prove to be extremely important factors in the values of their gaseous emissions and in the flue gas discharge stack, and have a significant effect on the health of humans, animals, as well as the environment.

To reduce dust emissions that include HCl and HF, filter bags or high-performance electrophiles are used, as shown in Figure 1. Modern technologies have also been adopted in a cement factory in Romania, which uses low-emission burners, and the grinding and burning processes have undergone extensive optimization operations, including through the selective non-catalytic reduction or addition of specific absorbers.

Compared to 1990, the dust emissions produced by the Romanian cement industry were reduced by about 95%, and the quantities of nitrogen oxide and sulfur dioxide decreased by 50%. In this range are also the levels of HCl and HF emissions [42].

## 4. Conclusions

Only through a unitary approach at a national level, through the elaboration of authorized methodological norms, as well as through guides for their implementation, can the best decisions be taken regarding the reduction of harmful gas emissions levels in the atmosphere.

The purpose of this paper was not to analyze the level of all the gaseous emissions at the cement plants in Romania (knowing very well the carbon effect through the level of CO_2_ emitted in the atmosphere, on the greenhouse effect and on global warming), but only of the gaseous compounds of chlorine and fluoride, which are also important for not only human health (and animals) but also for the vegetation and environment in which we live.

It is difficult to gather information on gaseous emissions when cement plants do not want to present them to the general public. The authors managed to collect some information from the data determined at the two factories analyzed through the specialized laboratories that make periodic field determinations. Unfortunately, it was not possible to collect information strictly related to the quantities of waste (by categories) during the periods and days when the determinations were made. These determinations are performed at least twice a year, according to current regulations.

The emissions of polluting gases containing HCl and HF are mainly due to the combustible materials used in clinker ovens. Although the levels of emissions for these polluting compounds have been below the allowed legal limits, they can be further reduced by using fuels with low emission potential or by an adequate processing thereof. Furthermore, the depollution systems used to discharge the flue gases into the atmosphere must be very efficient to further reduce the level of the polluting compounds and their retention in the filter elements.

Exposure of humans to contaminants like HCl and HF may result in many types of health damage ranging from relatively innocent symptoms such as skin eruption or nausea, to cancer or even death. Human health protection is generally considered a major protection target. The next step of the research will concern the analysis of the proposed approach from the human health point of view.

## Figures and Tables

**Figure 1 ijerph-17-01019-f001:**
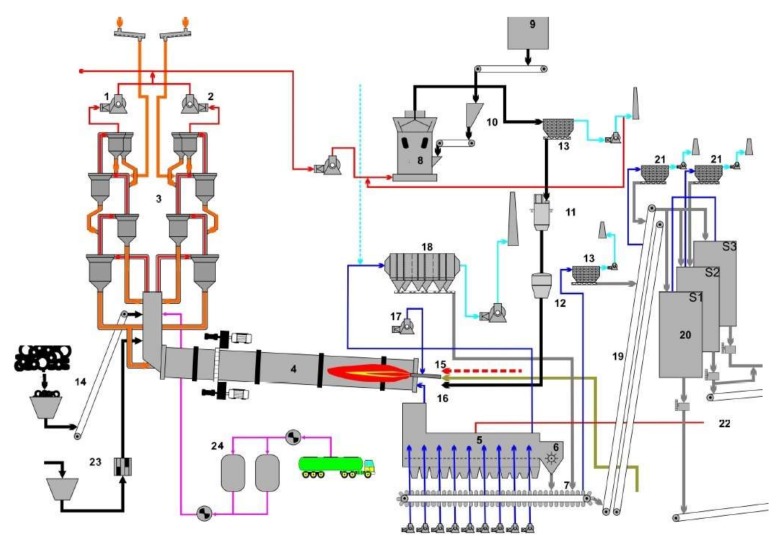
The scheme of the area where the cement factory is with a rotary cement kiln with clinker: 1. branch fan A; 2. branch fan B; 3. heat exchanger; 4. rotary cement kiln with clinker; 5. barbecue type cooler; 6. clinker crasher; 7.conveyers; 8. coal mill; 9. coal silo; 10. coal bunker; 11. milled coal silo; 12.proportioning system; 13. bag filter; 14. supply of used tires; 15.burner; 16.prefiring; 17. primary air fan; 18.electer filter; 19. conveyor belts; 20. clinker silos; 21. dust filters; 22. hot gas pipeline; 23. supply of oil sludge; and 24. NOx reduction installation.

**Figure 2 ijerph-17-01019-f002:**
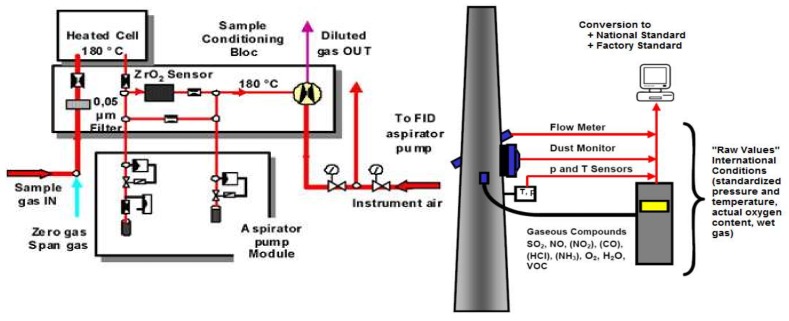
Simplified scheme and component elements of the sampling system for the clinker kiln emission evacuation tower.

**Figure 3 ijerph-17-01019-f003:**
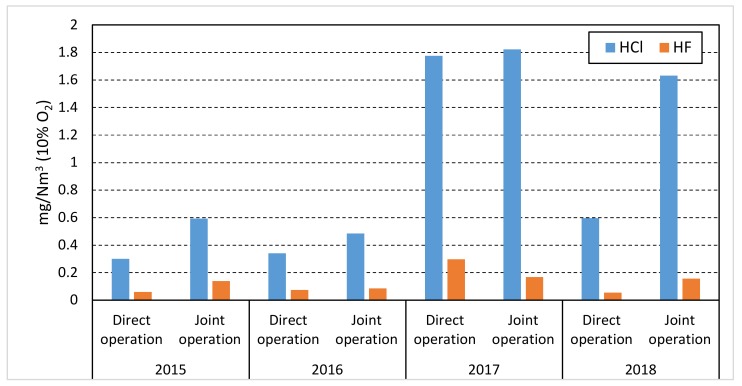
Evolution of HCl and HF emissions at the first analyzed cement factory.

**Table 1 ijerph-17-01019-t001:** Measurement results of HCl and HF of the rotary kiln at first cement factory, Romania, for 2015 and 2016.

Year	2015				2016			
Pollutant	HCl (Dry)		HF (Dry)		HCl (Dry)		HF (Dry)	
Working Mode	(mg/Nm^3^)	(mg/Nm^3^) 10% O_2_	(mg/Nm^3^)	(mg/Nm^3^) 10% O_2_	(mg/Nm^3^)	(mg/Nm^3^) 10% O_2_	(mg/Nm^3^)	(mg/Nm^3^) 10% O_2_
Direct operation *	0.255	0.229	0.059	0.053	0.436	0.387	0.084	0.074
	0.424	0.382	0.090	0.031	0.470	0.417	0.092	0.081
	0.325	0.293	0.059	0.048	0.250	0.222	0.078	0.069
	-	0.301	-	0.061	-	0.342	-	0.075
Joint operation **.	0.355	0.315	0.100	0.094	0.761	0.639	0.111	0.096
	0.515	0.485	0.184	0.173	0.214	0.179	0.034	0.029
	1.039	0.977	0.159	0.149	0.760	0.638	0.152	0.131
	-	0.592	-	0.139	-	0.485	-	0.085

* Direct operation—rotary kiln without raw meal mill; ** Joint operation—rotary kiln with raw meal mill.

**Table 2 ijerph-17-01019-t002:** Measurement results of HCl and HF of the rotary kiln at first cement factory, Romania, for 2017 and 2018.

Year	2017				2018			
Pollutant	HCl (Dry)		HF (Dry)		HCl (Dry)		HF (Dry)	
Working mode	(mg/Nm^3^)	(mg/Nm^3^) 10% O_2_	(mg/Nm^3^)	(mg/Nm^3^) 10% O_2_	(mg/Nm^3^)	(mg/Nm^3^) 10% O_2_	(mg/Nm^3^)	(mg/Nm^3^) 10% O_2_
Direct operation *	3.383	3.263	0.271	0.261	1.187	1.138	0.114	0.109
	0.991	0.956	0.311	0.300	0.371	0.355	0.039	0.037
	1.149	1.108	0.342	0.330	0.311	0.298	<0.019	<0.018
	-	1.776	-	0.297	-	0.597	-	<0.055
Joint operation **	0.834	0.811	0.081	0.079	2.628	2.981	0.157	0.178
	2.000	1.946	0.175	0.170	1.016	1.153	0.150	0.171
	2.785	2.710	0.262	0.255	0.670	0.760	0.105	0.119
	-	1.823	-	0.168	-	1.631	-	0.156

* Direct operation—rotary kiln without raw meal mill; ** Joint operation—rotary kiln with raw meal mill.

**Table 3 ijerph-17-01019-t003:** Measurement results of HCl and HF of the rotary kiln at the second factory for 2015.

Year	2015			
Pollutant	HCl (Dry)		HF (Dry)	
Working Mode	(mg/Nm^3^)	(mg/Nm^3^) 10% O_2_	(mg/Nm^3^)	(mg/Nm^3^) 10% O_2_
Branch A *	2.650	2.468	0.389	0.363
	2.898	2.699	0.521	0.485
	0.882	0.822	0.135	0.126
	-	1.996	-	0.325
Branch B **	0.277	0.258	0.078	0.073
	0.261	0.244	0.219	0.204
	0.502	0.468	0.076	0.071
	-	0.323	-	0.116

* Branch A, ** Branch B are indicated in Figure 1.

**Table 4 ijerph-17-01019-t004:** Measurement results of HCl and HF of the rotary kiln at the second factory for 2017 and 2018.

Year	2017				2018			
Pollutant	HCl (Dry)		HF (Dry)		HCl (Dry)		HF (Dry)	
Working Mode	(mg/Nm^3^)	(mg/Nm^3^) 10% O_2_	(mg/Nm^3^)	(mg/Nm^3^) 10% O_2_	(mg/Nm^3^)	(mg/Nm^3^) 10% O_2_	(mg/Nm3)	(mg/Nm^3^) 10% O_2_
Branch A *	0.251	0217	0.073	0.063	0.421	0.367	0.087	0.079
	0.503	0.416	0.072	0.059	0.567	0.489	0.067	0.058
	0.500	0.419	0.068	0.057	0.394	0.348	0.068	0.059
	-	0.351	-	0.060	-	0.402	-	< 0.065
Branch B **	0.345	0.299	0.069	0.060	0.805	0.701	0.145	0.136
	0,353	0.300	0.072	0.063	0.604	0.511	0.095	0.080
	0,301	0.276	0.073	0.067	1.933	1.775	0.100	0.091
	-	0.292	-	0.063	-	0.995	-	0.103

* Branch A, ** Branch B are indicated in Figure 1.

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
