# Peer review of "Emissions Control of Hydrochloric and Fluorhydric Acid in cement Factories from Romania"

_ijerph, 2020, doi:10.3390/ijerph17031019_

Round 1

Reviewer 1 Report

I am not a domain expert of cement production emissions. However, I want to comment on this paper, as an emission and pollution study for its fitness to the scope and interests to general readers to this journal. The current version is not suitable to be disposed on the journal because it fails to fit the journal scope to understand the relationship between emissions and public health and other wider effects of emissions. In the revised version, even if the authors do not add any new analyses, it is necessary to discuss the pollution effects on society, public, and other wider domains. Currently it fits better for a field-specific journal not this journal with broader readership. Line 15-16: The first sentence in the abstract is not comprehensible Line 31: For interests of journal readers, the paper needs to first discuss the emission (of different kinds) impacts and interactions with human societies, for example: https://link.springer.com/article/10.1007/s13280-019-01290-y https://www.sciencedirect.com/science/article/pii/S0959652617316815 https://www.sciencedirect.com/science/article/pii/S0959652619318104 Line 82: This short paragraph appears to be strange and not comprehensible. Line 129: Should this sentence be in the paper? Equations: all symbols need to be explained in the equations. Line 226: Should there be a comparison of standard worldwide or at least with other European countries? Discussions: This paper currently is way too short and lacks discussions for the implications of results and findings – both implications for the general public and policy implications, and carbon emissions with cement production is a very hot topic in the field now but totally ignored, for example: https://www.sciencedirect.com/science/article/pii/S0921344918304737 https://www.sciencedirect.com/science/article/pii/S0959652619302215

Reviewer 2 Report

In this manuscript, G. Voicu et al. analyzed the recent emission levels of HCl and HF in the Romania cement industry. They compared these values with the previous period, 2004~2013, and found that they were still in the legal range. Also, they point out that these pollutants are mainly generated from the combustible materials used in clinker ovens. However, the current version of the manuscript is incomplete because it does not contain any discussions. The authors should make proper discussions on what they have found. Besides, there are many typos in the manuscript; for example, HCL should be HCl and some formulas should be using a subscript. Therefore, I recommend reconsidering this manuscript after revised properly.

Reviewer 3 Report

The article is well written and structured.

Although they are widely known on line 144 (O, the first time they are used) The abbreviation VOC and FID should appear close to corresponding name.

On line 147 "a BRAVO M Plus constant flow sampling pump for non-kinetic sampling of HCL, NH3"

Is it possible that it should be written? "a BRAVO M Plus constant flow sampling pump for non-kinetic sampling of HCL and NH3".

The image quality of the equations is poor and the sizes are different.

An error is observed on line 196 "temperature 0oC, for".

In different lines 197, 201 .. O2 is not well written.

In line 207 Vtot must be written with its subscript.

in table 1. Review the superscript and subscript for example "Working mode (mg / Nm3)".

On line 239 please center the text.

The results shown have with them the regulatory limit values to better understand the results shown. No error bars are displayed in the results shown.

There is no justification for why there is an increase in 2017-2018 of the emissions levels of HCL and HF.

In the conclusions section it would be interesting to add some conclusions directly related to the work done to emphasize the importance of the data obtained.

Reviewer 4 Report

The manuscript may be accepted with minor revisions

In my opinion the Introduction can be summarized and many details may be organised in tables

Check the sentence at row 73 "As for chlorine, chlorines are minor additional...."

Table 2 is not cited in the text

The caption of table 3 has to be modified including the name of the second factory

The list of references may be updated, eliminating some dated citations

Round 2

Reviewer 1 Report

The revision has addressed reviewer concerns and can be published. 

Reviewer 2 Report

This is my second review on “Emissions control of Hydrochloric and Fluorhydric Acid in cement factories from Romania” submitted by G. Voicu et al. The authors revised the manuscript properly according to reviewers’ comments. The revised manuscript is now acceptable in the International Journal of Environmental Research and Public Health as it is.